# Differentially Expressed Genes Induced by Erythropoietin Receptor Overexpression in Rat Mammary Adenocarcinoma RAMA 37-28 Cells

**DOI:** 10.3390/ijms24108482

**Published:** 2023-05-09

**Authors:** Zuzana Tóthová, Martina Šemeláková, Katarína Bhide, Mangesh Bhide, Andrej Kováč, Petra Majerová, Monika Kvaková, Jana Štofilová, Zuzana Solárová, Peter Solár

**Affiliations:** 1Department of Medical Biology, Faculty of Medicine, P.J. Šafárik University in Košice, 04001 Košice, Slovakia; zuzana.tothova1@student.upjs.sk (Z.T.); martina.semelakova@upjs.sk (M.Š.); 2Laboratory of Biomedical Microbiology and Immunology, University of Veterinary Medicine and Pharmacy in Košice, 04001 Košice, Slovakia; katarina.bhide@gmail.com (K.B.); bhidemangesh@gmail.com (M.B.); 3Institute of Neuroimmunology, Slovak Academy of Sciences, 84510 Bratislava, Slovakia; andrej.kovac@savba.sk (A.K.); petra.majerova@savba.sk (P.M.); 4Department of Experimental Medicine, Faculty of Medicine, P.J. Šafárik University in Košice, 04001 Košice, Slovakia; monika.kvakova@upjs.sk (M.K.); jana.stofilova@upjs.sk (J.Š.); 5Department of Pharmacology, Faculty of Medicine, P.J. Šafárik University in Košice, 04001 Košice, Slovakia; zuzana.solarova@upjs.sk

**Keywords:** mammary adenocarcinoma, RAMA 37-28, erythropoietin receptor

## Abstract

The erythropoietin receptor (EPOR) is a transmembrane type I receptor with an essential role in the proliferation and differentiation of erythroid progenitors. Besides its function during erythropoiesis, EPOR is expressed and has protective effect in various non-hematopoietic tissues, including tumors. Currently, the advantageous aspect of EPOR related to different cellular events is still under scientific investigation. Besides its well-known effect on cell proliferation, apoptosis and differentiation, our integrative functional study revealed its possible associations with metabolic processes, transport of small molecules, signal transduction and tumorigenesis. Comparative transcriptome analysis (RNA-seq) identified 233 differentially expressed genes (DEGs) in EPOR overexpressed RAMA 37-28 cells compared to parental RAMA 37 cells, whereas 145 genes were downregulated and 88 upregulated. Of these, for example, *GPC4*, *RAP2C*, *STK26*, *ZFP955A*, *KIT*, *GAS6*, *PTPRF* and *CXCR4* were downregulated and *CDH13*, *NR0B1*, *OCM2*, *GPM6B*, *TM7SF3*, *PARVB*, *VEGFD* and *STAT5A* were upregulated. Surprisingly, two ephrin receptors, *EPHA4* and *EPHB3*, and *EFNB1* ligand were found to be upregulated as well. Our study is the first demonstrating robust differentially expressed genes evoked by simple EPOR overexpression without the addition of erythropoietin ligand in a manner which remains to be elucidated.

## 1. Introduction

EPOR cytokine receptor occurs in diverse types of cells, mainly erythropoietic progenitors and several types of non-hematopoietic tissues as well as cancer cells. In early erythroblasts, EPOR holds the position of regulator of cell size and cell cycle duration [1], and induces transcriptional reprogramming throughout their maturation [2]. EPOR and its tissue-protective effect was described in nervous system, retina, heart, kidneys, endothelium, muscle and bone tissues [3,4,5,6,7,8,9]. EPO/EPOR axis is also implied in regulation of energy metabolism, obesity and insulin response [10,11]. As a typical member of the type I cytokine receptor superfamily with no intrinsic kinase activity, the EPOR molecule consists of extracellular, transmembrane and intracellular regions. Part of the EPOR intracellular region named cytoplasmic box1 is constitutively associated with JAK2 kinases [12] and after binding of the EPO ligand, a cascade of phosphorylations is activated. Activation of the EPO/EPOR complex triggers signaling cascades JAK2/STAT5, PI3K/AKT, RAS/MAP and PKC signaling pathways [13]. It was reported that for successful signal transmission, EPOR translocates to membrane lipid microdomains along with JAK2, STAT5 and LYN kinase [14]. After these events, the signal translocates to the nucleus, where transcription of EPO responsive target genes is turned on. To date, the overall understanding of EPOR effects on the biological processes in the cells is still missing. Global transcriptome and proteome studies were formerly applied to examine EPO/EPOR effects during embryonic and adult erythropoiesis [15,16]. Lately, phospho-proteome analysis of erythroid progenitors detected 121 novel EPO/EPOR target proteins, including core regulators of metabolic processes aldolase and pyruvate dehydrogenase-α1, and also phosphorylation of 19 novel cytoskeletal targets [17]. Although EPOR signaling is well described in a variety of tissues and cells [13,14] studies dealing with EPOR-induced DEGs in malignant cells are completely missing. EPOR plays an important role in the progression of numerous cancers and can be used as a prognostic marker [18], but the precise mechanism of EPOR action on cancer cells needs to be further investigated. In breast cancer, its expression on the surface of tumor cells is associated with better survival, proliferation and invasiveness [19].

The objective of the present study was to assess the effects of EPOR on the whole transcriptome and proteome changes and interpret these results with regard to cell biological processes. For our purpose, we used rat mammary adenocarcinoma RAMA 37-28 cell line with stable EPOR overexpression, which is a subclone of benign non-invasive mammary rat cell line RAMA 37 [3]. Using RNA-seq and proteomics, a global picture of signaling events in RAMA 37-28 vs. RAMA 37 cells has been obtained, in which 233 DEGs were affected. Subsequently, the accuracy of the expression profile of 13 DEGs from RNA-seq result was verified using qRT-PCR. The main goal of our study was to reveal new candidate genes associated with EPOR expression and to categorize them according to their biological processes.

## 2. Results

### 2.1. Differentially Expressed Genes and Validation

We used RNA-seq analysis to understand the molecular changes occurring after the introduction and overexpression of human EPOR in the mammary adenocarcinoma cell line RAMA 37. RNA isolated from both the EPOR overexpressed clone RAMA 37-28 and the parental line RAMA 37 was evaluated for its quality. cDNA libraries were prepared from biological replicates that had an optimal fragment size between 150–300 nt. Sequencing yielded 10 million raw reads per sample of both the RAMA 37 and RAMA 37-28 lines. A total of 15,177 genes were mapped for each sample. Genes with a minimum mean logCPM (counts per million) of 3 were considered differentially expressed and included in the differential expression analysis. Genes with logFC (fold change) in the range above ±1.2 were included in the final list of differentially expressed genes (DEGs). Additionally, we checked the *p*-value for shortlisted DEGs and removed any gene with *p*-value > 0.01 (Appendix A).

A total of 233 genes were differentially expressed in RAMA 37-28 versus RAMA 37 cells (Appendix A). Among them, 145 were downregulated and 88 were upregulated. To validate the results obtained from RNA-seq, 13 DEGs were analyzed by qRT-PCR. The correlation of both results was carried out by calculating the Pearson correlation coefficient, the value of which was equal 0.943 (*p*-value = 1.3 × 10^−7^) (Figure 1). DEGs were divided based on GO biological processes using the peer-reviewed server Omnalysis (http://lbmi.uvlf.sk/omnalysis.html, accessed on 15 February 2023) (Figure 2).

### 2.2. DEGs Involved in Erythropoietin Signaling

Three genes are associated with erythropoietin signaling, of which two are upregulated and one is downregulated (GO biological processes “erythropoietin signaling”). *EPOR* overexpression evoked the upregulation of *STAT5A* and *LYN* and downregulation of *KIT*.

### 2.3. DEGs Involved in Proliferation

Forty genes associate with cell proliferation (GO biological processes “cell proliferation”). In this regard, 25 genes were downregulated and 15 genes were upregulated as a result of EPOR overexpression in RAMA 37-28 cells. Twelve downregulated genes *KIT* (logFC −7.93), *NPPB* (logFC −7.71), *SFRP2* (logFC −5.07), *SLURP1* (logFC −2.28), *XDH* (logFC −2.71), *NR4A1* (logFC −2.45), *CLDN1* (logFC −2.07), *SMAD3*, *IFT80*, *CDH3*, *ECM1* and *RGCC* and seven upregulated genes *CDH13* (logFC 8.01), *CXCL12* (logFC 2.28), *TGFB2* (logFC 2.04), *FST*, *FGFR2*, *STAT5A* and *SIX1* associate with negative regulation of epithelial cell proliferation. Furthermore, upregulated *CDKN1A* and *PAK1* can associate with negative regulation of vascular-associated smooth muscle cell proliferation and/or negative regulation of cell proliferation involved in contact inhibition (Figure 3a).

### 2.4. DEGs Involved in Apoptosis

Twenty-nine genes were categorized either in the regulation of apoptotic signaling pathway or simply involved in the apoptotic process. Of these, 15 were downregulated and 14 were upregulated (GO biological process “apoptosis”). The most downregulated *PTPRF* (logFC −5.67), *SFRP2* (logFC −5.07), *EGLN3* (logFC −2.91), *XDH* (logFC −2.71) and *NR4A1* (logFC −2.45) associate with negative regulation of both intrinsic and extrinsic apoptotic signaling pathways. Moreover, downregulated *EGLN3*, *NR4A1* and *SMAD3* can participate also in the inhibition of cysteine-type endopeptidase activity (involved in apoptotic process) and downregulated *ANGPT1* and *RGCC* correlate with diminishing of both epithelial and endothelial apoptosis. In spite of the proapoptotic (tumor suppressor) potential of upregulated genes *AKAP12* (logFC 3.36), *TGFB2* and *BTG2*, the group of overexpressed genes *CXCL12* (logFC 2.28), *ZFP385A* and *CDKN1A* associate with negative regulation of the intrinsic apoptotic signaling pathway in response to DNA damage by p53 class mediator (Figure 3a).

### 2.5. DEGs Involved in Cell Migration

Forty genes associate with migration, of which 23 are downregulated and 17 upregulated (GO biological processes “cell migration”). Despite EPOR-induced upregulated genes such as *TGFB2*, *PLK2*, *CTSH*, *AKAP12*, *PAK1*, *VEGFD* and *CXCL12*, the presence of downregulated genes *RAP2C* (logFC −12.01), *STK26* (logFC −11.04), *KIT* (logFC −7.93) and *SFRP2* (logFC −5.07) implies an association with negative regulation of RAMA 37-28 cell migration (Figure 3a).

### 2.6. DEGs Involved in Metabolic Processes

Sixty-two genes associate with metabolic processes, of which 38 are downregulated and 24 upregulated (GO biological processes “metabolic processes”). Many metabolic processes could be regulated as a result of EPOR-induced differentially expressed genes in RAMA 37-28 cells. The most significant seems to be altered lipid metabolism, in which downregulated *KIT* (logFC −7.93), *PMP22* and *CLN6* and upregulated *NR0B1* (logFC 7.70), *IGFBP7*, *THRB*, *STAT5A*, *ADM* and *CYP1B1* associate with steroid metabolic process (GO biological processes “metabolic processes”). While upregulated *LYN* and *RAB38* associate with the regulation of phospholipid and lipid metabolic processes, upregulated *THRB* could be directly involved in the regulation of triglyceride or cholesterol metabolic processes. In addition, *PTGS1*, *GSTA1*, *PDPN*, *DAGLA* and *CYP1B1* genes associate with icosanoid, unsaturated fatty acid and long-chain fatty acid metabolism. In relation to apoptosis, the following associations between the metabolic processes and DEGs appear to be important. The first one is between the metabolism of reactive oxygen species and both downregulated *XDH* and *SMAD3* and upregulated *DDAH1* (logFC 3.67), *CDKN1A* and *CYP1B1* genes. The second is between the metabolism of both xenobiotic and glutathione and the upregulated *GSTA1* gene, and the last one is between diminished phosphatidylserine metabolism and downregulated *PLSCR1* and *SERINC5* genes (Figure 3b).

### 2.7. DEGs Involved in Cell Differentiation

Eighty-four genes associate with cell differentiation, of which 50 are downregulated and 34 upregulated (GO biological processes “cell differentiation”). A large group of genes associates with the regulation of cell morphogenesis involved in cell differentiation including upregulated *OCM2* (logFC 7.08), *SEMA3D* (logFC 6.08), *EPHA4*, *CXCL12*, *PDPN*, *PAK1*, *EPHB3*, *LYN* and *GPRC5B* and downregulated *CXCR4* (logFC −7.30), *FBLN1* (logFC −7.08), *PTPRF* (logFC −5.97), *ALK* (logFC −5.57), *LPAR1*, *DOCK1*, *STMN2*, *SHOX2*, *STAU2* and *SYT1* and *SEMA2F* (GO biological processes “cell differentiation”). Upregulated genes *CDKN1A*, *SHROOM3* and *FGFR2* associate also with columnar and/or cuboidal epithelial cell differentiation. Indeed, many genes evoked by EPOR overexpression correlate with differentiation of muscle, neuronal and other cells, e.g., *TSHZ3* (logFC −11.34), *ADRA1B* (logFC −9.27), *TMEM119*, *KIT*, *PMP2*, *NPPB*, *KRT8*, *SORBS2*, *RAMP2*, *SIX1*, *ADM* and *DLL1*, but their role in the differentiation of RAMA 37-28 cells remains unclear and requires additional studies (Figure 3b).

### 2.8. DEGs Involved in Signal Transduction

Ninety-two genes associate with signal transduction, of which 55 are downregulated and 37 upregulated (GO biological processes “signal transduction”). Most signaling pathways in RAMA 37-28 cells appear to be downregulated based on changes in the expression of categorized genes (GO biological processes “signal transduction”). In this case, both small GTPase-mediated signal transduction via downregulated genes *RAP2C* (logFC −12.01), *ADRA1B* (logFC −9.27), *CCDC88C* (logFC −5.63), *RGL1*, *WASF1*, *RHEBL1*, *LPAR1*, *DOCK1* and *RERG* and Ras signal transduction via downregulated *SFRP2* (logFC −5.07), *ANKRD6*, *SMAD3*, *BAMBI* and *CDH3* are inhibited. Except for upregulated *STAT5A*, cytokine-mediated signaling is also downregulated via reduced expression of *KIT* (logFC −7.93), *GAS6* (logFC −7.53), *CXCR4* (logFC −7.30), *PTPRF* (logFC −5.97), *IL17RC*, *KRT8*, *ANGPT1* and *ECM1* genes. Finally, the intrinsic apoptotic signaling pathway in response to DNA damage is also negatively regulated based on downregulated *PLSCR* and *ATM* and upregulated *CDKN1A*, *CYP1B1*, *CXCL12* and *ZFP385A* genes. On the contrary, the combination of upregulated *CTNND2*, *GPRC5B* and *FGFR2* and downregulated *CCDC88C* and *ANKRD6* associate with positive regulation of Wnt signaling. Interestingly, ephrin receptor signaling pathway with three upregulated *EFNB*, *EPHA*, *EPHB3* is also positively regulated in RAMA 37-28 cells (Figure 3c).

### 2.9. DEGs Involved in Cell Transport

According to GO biological processes “transport of small molecules”, 58 genes were found, of which 41 are downregulated and 17 upregulated. The majority of EPOR-evoked genes in RAMA 37-28 cells associate with the regulation of transmembrane protein, hormone and ion transport. While in protein transport, *PRR5L*, *MMP13*, *CD14*, *RGCC* and *GAS6* were downregulated and *TM7SF3*, *TGFB2* and PAK1 were upregulated; in hormone transport, *C1QTNF1*, *ICA1*, *PTGER3*, *PER2*, *LTBP4*, *FOXD1*, *CPE* and *CRYM* were downregulated and *LYN*, *FST* and *ADM* upregulated. The most significantly downregulated genes associated with protein and hormone transport were *FOXD1* (logFC −9.13) and *GAS6* (logFC −7.53), respectively, whereas the *TM7SF3* gene was significantly upregulated (logFC 4.34) in both transports. In the case of ion transmembrane transporter activity regulation, downregulated *PTGER3*, *GPM6B*, *KCNT2*, *WNK4*, *KCNK5* and *SYT1* with most significant *RAMP1* (logFC −8.68) and *CXCR4* (logFC −7.31) and upregulated *FHL1*, *PTAFR*, *CXCL12*, *TGFB2* and *LYN* genes were categorised. The four abovementioned *TGFB2*, *GPM6B*, *PTGER3* and *WNK4* genes associate also with negative regulation of calcium transmembrane transport (Figure 3c).

### 2.10. DEGs Involved in Tumorigenesis

Seventy genes associate with tumorigenesis, of which 47 are downregulated and 23 upregulated (GO biological processes “tumorigenesis”). Among them are 15 oncogenes and 10 tumor suppressors which can contribute to the development of a malignant phenotype of RAMA 37-28 cells. Moreover, overexpression of subset of genes was found to be associated with breast cancer tumorigenesis (*ID4*, *SOX5*, *ITM2A*, *SNCG*, *EPHA4*, *NTS*, *CYP1B1*, *FGFR2*, *ENPEP*), whereas *ID4*, *SEMA3F*, *FOXD*, *KCNK5*, *WNK4* and *ECM1* genes may play a role in the chemoresistance of cancer cells. Interestingly, upregulated DEGs like *SNCG* (logFC 9.95), *CPNE8* (logFC 7.18), *GRIA3* (logFC 4.08), *SOX5* (logFC 3.94) and *CRISP3* (logFC 3.44) representing the highest log fold changes were involved in tumor cell metastasis and invasiveness. Transcriptome analysis revealed also upregulated *RADX* (logFC 9.92) and downregulated *MLH6* (logFC −2.00), *NPAT*, *ATM* and *SMARCA5* genes associated with DNA damage response (Figure 3c).

### 2.11. Proteomic Analysis and Validation by Western Blot

Proteome analysis of more than 5000 peptides confirmed the altered expression of *OCM2* (logFC 7.43) and *SNCG* (logFC 2.08) genes from the transcriptome analysis, but also revealed new DEGs and/or proteins such as ALDH6 (logFC 5.51), SPRR1B (logFC 2.64), ZYX (logFC 2.28), CTPB (logFC 4.62), HPRT (logFC −5.31) and hypothetical protein MJ0443 (logFC −3.51) (Appendix A). To validate the results obtained from proteomic analysis, five representative proteins were analyzed using Western blot. Results obtained from both techniques were consistent (Appendix A). EPOR-induced changes were confirmed by using specific siRNA designed against EPOR mRNA. In addition, 48 h silencing of EPOR did not abolish the expression of selected STAT5A and OCM2 genes; however, it significantly reduced their protein levels. On the contrary, EPOR silencing did not increase the expression of downregulated HPRT (Figure 4).

## 3. Discussion

EPOR was discovered in 1989 [20] and has been intensively studied for over 30 years. Activation of EPO-EPOR cascades during erythropoiesis results in transcriptional activation of proliferation and differentiation [2], while in non-hematopoietic tissues, anti-apoptotic and protective mechanism of EPOR signaling is described [21,22,23]. The expression of EPOR has been demonstrated in a panel of 29 tumor cell lines [24] and may be related to increased resistance of cancer cells to various therapies. In the current study, we describe EPOR as a causative factor of extensive expression changes referred to 233 DEGs in RAMA 37-28 cell line, affecting numerous biological functions and resulting in the origin of a new cell identity. The genes obtained from RNA-seq have been processed and clustered into particular biological categories (GO biological processes). In our previous studies, we have shown that RAMA 37-28 cells have lower proliferation capacity compared to RAMA 37 cells, as a result of EPOR overexpression [25,26] (Appendix A). Currently, we used xCelligence screening to show faster proliferation of RAMA 37-28 cells under conditions of paclitaxel added, whereas EPOR silencing suppressed such an effect (Appendix A).

According to our transcriptome analysis, EPOR overexpression in RAMA 37-28 cell line results in 40 DEGs associated with cell proliferation. Indeed, 12 downregulated genes such as *KIT*, *NPPB*, *SFRP2*, *SLURP1*, *XDH*, *NR4A1*, *CLDN1*, *SMAD3*, *IFT80*, *CDH3*, *ECM1* and *RGCC* and seven upregulated genes, *CDH13*, *CXCL12*, *TGFB2*, *FST*, *FGFR2*, *STAT5A*, *SIX1*, are involved in the inhibition of epithelial cell proliferation. In this regard, STAT5A, as a member of JAK2/STAT5 signaling cascade regulated by EPO/EPOR activation, is identified as a chemoresistance inducer and the regulator of ABCB1 transporter protein [27] with the ability to stabilize also heterochromatin structure followed by the inhibition of cell growth [28]. We demonstrate EPOR-induced overexpression of STAT5A, whose direct relationship (without EPO addition) using siRNA against EPOR was confirmed. Upregulated TGFB itself encodes the transforming growth factor beta family of cytokines which functions in proliferation, differentiation, adhesion and migration in many cell types [29]. TGFB receptors, moreover, activate both SMAD-dependent and independent pathways that not only regulate SMAD signaling, but also allow SMAD-independent TGFB responses [30]. The data of Livitsanou et al. [31] demonstrated a novel mechanism of TGFB/SMAD signaling modulation by the small GTPase RHOB and show that TGFB/RHOB signaling cross talk affects the nuclear and cytoplasmic responses to TGFB in opposite ways. The slower proliferation of our RAMA 37-28 cells could therefore be explained by the downregulation of *SMAD3* and the upregulation of *TGBF2* and *RHOB* genes. Furthermore, Marlow et al. [32] showed that RHOB has divergent downstream signaling partners, which are dependent on the HDAC isoform that is inhibited. When RHOB upregulates only P21 using a class I HDACi (romidepsin), cells undergo cytostasis. When RHOB upregulates BIM using class II/(I) HDACi (belinostat or vorinostat), apoptosis occurs. Combinatorial synergy with paclitaxel is dependent upon RHOB and BIM while upregulation of RHOB and only P21 blocks synergy. Interestingly, RAMA 37-28 cells which demonstrate paclitaxel resistance compared to RAMA 37 cells [26] reveal overexpression of both *RHOB* and *CDKN1A* genes. In search of mechanisms responsible for the proliferative changes of RAMA 37-28 cells, we emphasize the overexpression of *IGFBP7* and simultaneous downregulation of *NPPB* genes. Indeed, it has been reported that IGFBP7, involved in p53-dependent growth suppression of lung and colorectal tumors [33,34], decreased the production of NPPB during viral tumorigenesis [35], which points to the conditional expression of some DEGs. Moreover, overexpressed P21 activated *PAK1* may also be involved in the negative regulation of cell proliferation involved in the contact inhibition together with downregulated *CLDN-1*, whose lower expression is either associated with cancer progression (invasion) or improved survival of cancer patients [36]. Highly downregulated proto-oncogene and receptor tyrosine kinase KIT itself might be another important factor contributing to both neoplastic breast epithelium transformation and inhibition of cell migration which were already demonstrated by Janostiak et al. [37].

Furthermore, we demonstrate the upregulation of ephrin receptors (*EPHA4*, *EPHB3*) and ligand (*EFNB1*) which can associate with a slowdown in RAMA 37-28 proliferation. While EPHA4 signaling promotes mesenchymal-to-epithelial transition during morphogenesis in zebra fish embryo development [38], it was reported that overexpression of EPHB3 enhances cell–cell contacts and inhibits the growth in HT-29 human colon cancer cells [39]. EFNB1 plays important role in cell adhesion in healthy tissues [40], while in tumors, it inhibits cell proliferation and adhesion [41].

Interestingly, overexpression of EPOR is associated with the emergence of a malignant phenotype in RAMA 37-28 cell line [42]. In our previous studies, overexpression of EPOR in RAMA 37-28 breast cancer cells altered the sensitivity of these cells to tamoxifen and paclitaxel via mechanism related to prolonged activation of AKT and ERK1/2 pathways, respectively [25,26]. Acquisition of chemotherapy-resistant cancer phenotype of RAMA 37-28 cells may also associate with the overexpression of widely described cell cycle regulator *CDKN1A* (*P21^Waf1/Cip1^*) [43], which can induce cell cycle arrest, followed by reduction of cancer cell growth rate. Indeed, cytoplasmic P21 phosphorylated by AKT increases cell survival and contributes to taxol resistance in glioblastoma cells [44], cisplatin resistance in testicular and ovarian cancer [45,46], doxorubicin resistance in triple-negative breast cancer cells SUM159 [47] and failure of paclitaxel treatment in human nasal squamous carcinoma RPMI-2650 [48]. In addition, P21 mediates 5-fluorouracil resistance in colorectal cancer cells [49] and is associated with poor response to tamoxifen in MCF7 breast cancer cells [50]. *CDKN1A* overexpression drives cells to acquire a more aggressive phenotype that is capable of escaping cell block, senescence and apoptosis [43]. Equally important in the resistance of RAMA 37-28 cells to tamoxifen alone is *PAK1*, whose overexpression and the role in tamoxifen resistance of breast cancer patients has already been described [51]. Besides this, *ID4*, *SEMA3F*, *FOXD*, *KCNK5*, *WNK4* and *ECM1* associate with chemoresistance origin and *SOX5*, *ITM2A*, *SNCG*, *EPHA4*, *NTS*, *FGFR2* and *ENPEP* associate moreover with breast cancer tumorigenesis. The most significantly upregulated DEGs such as *SNCG*, *CPNE8*, *GRIA3*, *SOX5* and *CRISP3* can be involved in metastasis and invasivity of cancer cells as well. In this regard, *SNCG* is overexpressed in human infiltrating breast carcinomas and promotes metastasis [52], while it is undetectable in normal and benign breast tissues [53]. Tian et al. [54] indicated a possible relationship between SNCG and P21 and their relation to radioresistance. We show that EPOR evoked upregulation of SNCG (both mRNA and protein levels) in RAMA 37-28 cells and suppose that the overexpression of both *SNCG* and *P21* genes could play a significant role in the chemoresistance phenotype of RAMA 37-28 cells.

The sensitivity of a tumor cell to therapy can also be affected by changes in the cell’s metabolism [55]. Our transcriptome analysis reveals 62 DEGs associated with such a biological property and genes such as *XDH*, *CYP1B1*, *GSTA1*, *PLSCR1* and *DDAH1* are definitely genes worth mentioning. In addition, proteomic analysis enriched this category with ALDH6 and HPRT1 proteins. The expression of *XDH* inversely correlates with the expression of cancer stem cell–related genes, such as *CD44* or *CD133*, and their downregulation promotes TGFβ signaling in hepatocellular carcinoma [56]. It is noteworthy that knocking-down or inhibiting *XDH* results in development and progression of hepatocellular HCC and promotes migration and invasion but not proliferation of these cells, while the changes themselves are dependent on increased activation of TGFB2/SMAD2/3 signaling pathway [56]. It seems that low activity of *XDH* provides a selective privilege to cancer cells and may also contribute to neoplastic differentiation of RAMA 37-28 cells. However, the main biological effects of TGFβ are inhibition or promotion of proliferation, differentiation, apoptosis, cell dormancy, autophagy and cellular senescence; reduced levels of *SMAD3* in RAMA 37-28 cells can suppress the function of TGF-β-induced expression of tumor suppressor genes. So, downregulation of *TGFB2* results in the expression of anti-apoptotic proteins BCL2 and BCL-W, and enhanced cancer cell survival, which confers platinum resistance in NSCLC and 5-FU resistance in CRC cells, respectively [57,58]. A similar relationship to resistance to cisplatin and paclitaxel chemotherapy was described in the case of *GSTA1* and *CYP1B1* genes, respectively [59,60]. Although both genes are overexpressed in RAMA 37-28 cells, the *CYP1B1* could be one of the mechanisms explaining the resistance phenotype of RAMA 37-28 cells to paclitaxel [26]. The sensitivity and/or proliferative capacity of tumor cells can also be affected by phospholipid scramblase 1 (*PLSCR1*) which is involved in several processes including phosphatidylserine exposure on an apoptotic cell surface. Its upregulation promoted cell proliferation, invasion and migration, while its downregulation inhibited these effects [61]. On the contrary, downregulation of *PLSCR1* expression, which is also seen in RAMA 37-28 cells, significantly inhibited the proliferation, adhesion, migration and invasion of cancer cells [62]. Another important gene that is involved in cellular metabolism and at the same time plays an important role in the altered phenotype of the tumor cell is *DDAH1*. The encoded enzyme plays a role in NO generation by regulating cellular concentrations of methylarginines, which in turn inhibits NO synthase activity. While endothelium-derived NO stimulates angiogenesis through the inhibition of apoptosis [63], in cancer cells, the roles of NO are diverse, and might have dual pro- and anti-tumor effects depending on local concentration [64]. Finally, the upregulated ALDH6 enzyme confirmed in RAMA 37-28 cells by proteome analysis may also play a significant role in the resistance [65] of our cells, while the role of our downregulated HPRT1 in relation to the published results [66] is questionable.

It is well known that the characteristic attribute of cancer cells is alteration of cell adhesion and/or extracellular matrix organization (ECM). According to DEGs identified by transcriptome analysis of RAMA 37-28 cells compared to RAMA 37 control, we suppose the loss of cell adhesion ability of RAMA 37-28 cells via plasma membrane adhesion molecules which are closely related to strong downregulation of morphogen and tumor suppressor glypican 4 (*GPC4*). The effect of *GPC4* downregulation was earlier demonstrated in relation to breast tumor progression [67], as well as in healthy tissues, to disruption of epithelial integrity and tight junction organization [68]. GPCs have been shown as well-known and accepted cell surface coreceptors for growth factors such as Wnt/β-catenin, FGF, IGF, VEGF and TGFB and matrix modifying enzymes in many cancer cells, thus being involved in control of tumor growth, metastasis, angiogenesis and ECM [69,70,71]. Moreover, Varma et al. [72] showed that *GPC4* mRNA level was downregulated in oxaliplatin-resistant ovarian carcinoma cell line A2780/C10. We demonstrate expression changes also in Frizzled-related protein 2 (*SFRP2*), which is secreted and incorporated into ECM of the normal and tumor cells. Its association with the fibronectin–integrin protein complex, promotion of cell adhesion and/or transformation of normal mammary epithelial cells into a tumor was already described earlier [73]. *SFRP2* was downregulated in radiotherapy-treated glioma patients, and low *SFRP2* expression was correlated with advanced tumor stage and poor prognosis. CRISP/Cas9-mediated *SFRP2* knockdown promoted soft agar colony formation, cancer stemness and radioresistance of glioma cells, while increased *SFRP2* expression exhibited opposite effects. Moreover, *SFRP2* knockdown activated Wnt/β-catenin signaling in glioma cell lines, while overexpression of *SFRP2* inhibited Wnt/β-catenin activation [74]. In the case of ECM assembly, downregulated *LAMB3*, *SMAD3*, *RGCC* and *GAS6* and upregulated *GPM6B* were categorized. On the other hand, downregulated *MMP13* and *CST3* and upregulated *TGFB2* and *PDPN* correlate with ECM disassembly. Interestingly, upregulated *FBN1* associates with sequestering of *TGFB* in ECM. Many more DEGs of RAMA 37-28 cells are related to ECM remodeling and changes of cell adhesion, such as upregulated *CYR61*, *MMP10*, *NOV*, *CTGF*, *TMEM47*, *CDH13*, *CTNND2*, *PAK1* and *LOXL1* and downregulated *FBLN1*, *CLDN1*, *ECM1*, *TENM2*, *RGD1561161*, *COL16a1*, *COLA3*, *SGCE*, *THBS2*, *PTPRF*, *MEPE*, *C1QTNF1*, *PLEKHA2*, *PCSK6*, *FLRT2* and *GPM6A*.

Not only alterations in cell growth and proliferation but also different morphology in RAMA 37-28 cells compared to parental RAMA 37 ones was observed [25,26]. Although the molecular basis of such difference has not been fully described, actually, 84 genes out of 233 associate with cell differentiation of RAMA 37-28 cells. The most significantly overexpressed gene in this category, both at the level of mRNA and the protein itself, is high-affinity calcium ion-binding protein (*OCM2*), which belongs to the superfamily of calmodulin proteins. We conclude and show for the first time that the expression of the *OCM2* gene which associates with positive regulation of cell morphogenesis involved in differentiation (GO biological processes “cell differentiation”) is directly related to the expression of EPOR. This effect was confirmed by siRNA against EPOR. Indeed, high expression of *OCM2* was demonstrated earlier in rat cancer cell lines undergoing neoplastic transformation [75,76]. Besides *OCM2*, 19 DEGs participate directly in the morphogenesis process. Thus, we demonstrate several upregulated DEGs related to columnar or cuboidal epithelial differentiation, namely, *P21*, *SHROOM3* and *FGFR2*. A large body of evidence suggests that *P21* also plays an important role in the differentiation of various normal or malignant cells and tissues, but its effect is dependent on the cell type and the stage of differentiation. Interestingly, *P21* has a positive role in differentiation in most studies, partly by inhibiting apoptosis, which promotes cell survival [77]. The STAT5A protein itself plays a key role both in the positive regulation of myeloid cell differentiation and gamma-delta T cell differentiation, as well as in the negative regulation of erythrocyte differentiation, while its role in the differentiation of RAMA 37-28 cells remains unclear.

## 4. Materials and Methods

### 4.1. Culture of RAMA Cells

Rat mammary adenocarcinoma cell lines RAMA 37 and RAMA 37-28 were kindly provided by Dr. Mohamed El-Tanani (Center of Cancer Research and Cell Biology, Queens University Belfast, Belfast, UK). RAMA 37-28 cell line is a clone of parental RAMA 37 cells, with stabile overexpression of human EPOR using an expression vector pcDNA3.1/V5-His-TOPO and selection antibiotic Geneticin (Invitrogen, Carlsbad, CA, USA) (1 mg/mL). RAMA 37 and RAMA 37-28 cells were cultured in RPMI-1640 medium (Biosera, Cholet, France) supplemented by antibiotics (antibiotic-antimycotic solution 100×, Sigma-Aldrich, Darmstadt, Germany) and 10% of FCS (fetal calf serum) (Gibco, Carlsbad, CA, USA) in the presence of 5% CO_2_ in a humidified atmosphere at 37 °C. The 5 × 10^6^ cells were plated onto a 75 cm^2^ flask; the culture reached 70–90% confluency in 2–3 days and was ready to split or harvest for experiments. To determine the linear range of each assay, six cell densities ranging from 50–10,000 cells/well were plated into sterile 96-well plates and incubated for 24, 48 or 72 h.

### 4.2. Preparation of the Library

A quantity of 250 ng of RNA was reverse transcribed to first strand cDNA using oligodT primers and QuantSeq. 3′ mRNA-Seq Library Prep Kit (Lexogen, Wien, Austria). RNA template was removed using RNA removal solution (RS buffer) (Lexogen, Wien, Austria) and the second strand was synthesized using random hexamer primer with Illumina-compatible linker sequences at its 5′ end. The dsDNA libraries were purified using magnetic beads and then amplified by PCR using specific single indexing i7 primers with the aim of adding adapter sequences important for cluster generation and to generate DNA for quality control and sequencing. PCR Add-on kit for Illumina (Lexogen, Wien, Austria) was used for the determination of 20 PCR cycles required for RAMA cells. Later, magnetic beads of the kit were used for purification of amplified libraries, which were checked for the length of the fragments by fragment analyzer.

### 4.3. Sequencing and Data Analysis

cDNA libraries were sequenced using Illumina NextSeq, single-end 75 bp, with 8 million reads per sample. Fastq files were processed and aligned to reference genome (Rattus norvegicus, mRatBN7.2) using STAR aligner (STAR V 2.5.2b, (https://github.com/alexdobin/STAR/releases/tag/2.5.2b, accessed on 20 February 2023). The pre-processing included adaptor trimming and removal of recommended initial 10 bases. Reads were counted in STAR V 2.5.2b. to perform differential gene expression analysis edgeR (open source R package version 3.12 was used: https://bioconductor.org/packages/release/bioc/html/edgeR.html, accessed on 20 February 2023). The low read count with less than 3 CPM (count per million) was filtered out using the filterByExp function of edgeR package. The identification of differentially expressed gene (DEGs) was accomplished by using glmTreat and glmQLFit (quasi-likelihood, QL) functions of edgeR in R package considering log fold change (logFC) values beyond ±1.2 and FDR less than 0.05. The logical relation of DEGs between the challenged RAMA cells (RAMA 37-28 vs. RAMA 37) was calculated using Excel (MS office) and Venn diagrams were constructed. To categorize DEGs groups into GO biological processes and to construct heat maps https://reactome.org/ (accessed on 20 February 2023) and http://www.heatmapper.ca/expression/ (accessed on 20 February 2023), respectively, were used.

### 4.4. Validation of DEGs by qRT-PCR

Reverse transcription (RT) was performed at 37 °C in a 20 μL volume using 1 μg of the total RNA, 10 mM random hexamer primer and 200U M-MLV reverse transcriptase (Invitrogen, Carlsbad, CA, USA) according to the manufacturer’s instructions. Briefly, 1 μg of RNA and 100 pM of random hexamers were mixed and incubated 5 min at 65 °C. Subsequently, 4 μL of 5× reaction buffer, 2 μL dNTP (10 mM), 1 μL RevertAid reverse transcriptase (200 U) (Thermo Fisher Scientific, Waltham, MA, USA) and 0.5 μL RiboLock RNase inhibitor (20U) (Thermo Fisher Scientific, Waltham, MA, USA) were added. The reaction mixture was incubated for 10 min at 25 °C and 1 h at 42 °C, followed by 70 °C for 10 min. A set of DEGs significantly up and downregulated in RNA-seq (23 genes–upregulated and downregulated) were selected for qRT-PCR. Primers used in qRT-PCR were designed using Geneious Pro software 2020 (Biomatters, San Francisco, CA, USA) (Appendix A). Reaction mix of qRT-PCR was composed of 1 µg of cDNA, 1× Maxima SYBR Green/ROX qPCR Master Mix (2x) (Thermo Fisher Scientific, Waltham, MA, USA), gene-specific primers (10 pM each) and RNase free water up to total volume of 20 μL. Each reaction was performed in triplicate. Amplification cycles were as follows: 10 min at 95 °C, 40 cycles [15 s at 95 °C, 30 s at 60 °C, 30 s at 72 °C], 5 min at 72 °C; melting curve 54 °C to 95 °C–0.3% temperature increment/s (Eco™ qRT-PCR System, CA; Eco™ Software v4.1.2.0). The gene expression (ΔΔCt) was normalized to β–actin (house-keeping gene). ΔΔCt values were converted to logFC (https://goldbio.com/qpcr-and-rtqpcr-analysis-tool, accessed on 27 February 2023). The correlation of DEGs values obtained from RNA-seq and qRT-PCR was determined by calculating the Pearson correlation coefficient (r^2^). Correlation plots and Pearson correlation were performed by Prism 9 software (Graphpad, Boston, MA, USA).

### 4.5. RNA Isolation, Reverse Transcription and PCR Analysis

After incubation of RAMA cells in 6-well plates, mRNA was isolated using TRIzol^®^ reagent (Invitrogen, Carlsbad, CA, USA) according to the manufacturer’s instructions. DNaseI treatment was essentially incorporated during RNA isolation. Total RNA concentration and purity (OD 260/280 and 260/230 ratio) was measured using Nanodrop (Thermo Fisher Scientific, Waltham, MA, USA). The RNA was stored at −80 °C. Reverse transcription was performed according to the qRT-PCR method mentioned above.

### 4.6. Isolation, Purification of the Proteins and Western Blot Analysis

RAMA cells cultivated in 6-well plates were lysed in (lysis buffer FNN0011, Thermo Fisher Scientific, Waltham, MA, USA) and 1× protease inhibitor cocktail (PIC P8340, Sigma-Aldrich, Darmstadt, Germany), followed by sonication on ice (2 cycles; 30-s pulses, 100% amplitude). The cell lysates were sonicated for 15 s at 30% power of the BANDELIN SONOPULS HD2070 (BANDELIN electronic, Berlin, Germany) on ice and centrifuged for 10 min/13,000× *g* at 4 °C. Protein concentration was determined using a detergent-compatible protein assay (Bio-Rad Laboratories, Hercules, CA, USA). Equal protein amounts (30 μg) supplemented with 0.01% bromphenol blue, 1% 2-mercaptoethanol, 0.4% SDS and 5% glycerol were then separated with 10% SDS-polyacrylamide gel (Acrylamide/Bis Solution, 37.5:1) and transferred (dry transfer, 10 min) onto a NC membrane (Bio-Rad Laboratories, Hercules, CA, USA) by dry transfer (iBlot 2 Gel Transfer Device, Thermo Fisher Scientific, Waltham, MA, USA). After 1 h of membrane blocking at RT in 5% non-fat milk (20 mM Tris–HCl, pH 7.6, 150 mM NaCl, 0.1 % TWEEN 20, pH 7.4), the NC membrane blots were incubated overnight at 4 °C with primary antibody anti–Oncomodulin (Polyclonal Antibody PA5-115689 R&D); anti–Glypican 4 (Polyclonal Antibody, PA5-115301); anti–HPRT1 (Recombinant Rabbit Monoclonal Antibody, JU03-26); anti–gamma Synuclein (Recombinant Rabbit Monoclonal Antibody (JM90-32, MA5-32748); anti-STAT5 (ab 230670). After 20 min washing in wash buffer, the membranes were incubated with appropriate horseradish peroxidase-conjugated secondary antibody for 1 h at RT (Goat anti-rabbit IgG 1:2000, 31461 Thermo Fisher Scientific, Waltham, MA, USA Rabbit anti-mouse IgG 1:5000, Santa Cruz, Dallas, TX, USA). Detection of antibody reactivity was performed using a Pierce ECL Western Blotting Substrate (Pierce, Thermo Fisher Scientific, Waltham, MA, USA) and visualised by bio-imaging system (DNR MF Chemibis 2.0, Neve Yamin, Israel). Equal sample loading was verified by immunodetection of anti- β-actin Monoclonal Antibody AC-15 (MA1-91399). The pictures were scanned with GS-800 Calibrated Densitometer and the quantification was performed using Image J software version 1.52 (NIH; National Institutes of Health, Bethesda, MD, USA).

### 4.7. Proteomics Analysis

Total proteins (100 μg) were reduced with 10 mM dithiothreitol in 100 mM ammonium bicarbonate at 56 °C for 30 min. Alkylation was performed with 100 μL of 50 mM iodoacetamide in 100 mM ammonium bicarbonate in the dark for 30 min. Proteins were digested with trypsin in ratio 1:100 at 37 °C, overnight. Aliquots of purified complex peptide mixtures of 100 ng were separated using Acquity M-Class UHPLC (Waters, Etten-Leur, The Netherlands). Samples were loaded onto the nanoEase Symmetry C18 trap column (25 mm length, 180 μm diameter, 5 μm particles size). After 2 min of desalting/concentration by 1% acetonitrile containing 0.1% formic acid at a flow rate 8 μL/min, peptides were introduced to the nanoEase HSS T3 C18 analytical column (100 mm length, 75 μm diameter, 1.8 μm particle size). For the thorough separation, a 90 min gradient of 5–35% acetonitrile with 0.1% formic acid was applied at a flow rate of 300 nL/min. The samples were nanosprayed (3.1 kV capillary voltage) to the quadrupole time-of-flight mass spectrometer Synapt G2-Si with ion mobility option (Waters, Etten-Leur, The Netherlands). Spectra were recorded in a data-independent manner in high definition MSE mode. Ions with 50–2000 m/z were detected in both channels, with a 1 s spectral acquisition scan rate. Spectra were preprocessed with the Compression and Archival Tool 1.0 (Waters, Etten-Leur, The Netherlands) to reduce noise, removing ion counts below 15.

### 4.8. Silencing of EPOR Gene Expression by siRNA

For silence of EPOR expression in RAMA 37-28 cells, we used siRNAs (siRNA Dharma SMARTpool ON-TARGETplus human EPOR siRNA, GE HealthCare, PerkinElmer Holdings Inc., Lafayette, CO, USA), ntRNAs (negative controls to siRNA Dharma ON-TARGETplus–non-targeting) and transfect reagent DharmaFECT 1 (GE HealthCare, PerkinElmer Holdings Inc., Lafayette, CO, USA) following the manufacturer’s instructions. Transfection reagents were stable and none of the tested conditions significantly affected cell toxicity. RAMA 37-28 cells were cultured in 6-well plates (500,000 cells/well) in RPMI media without antibiotics, supplemented by 10% of FBS and incubated for 24 h at 37 °C in the presence of 5% CO_2_. Subsequently, cells were cultured in plates with changed fresh media without antibiotics, and were supplemented by 10% of FCS; the experimental groups were enriched by siRNA (14 µM) or ntRNA (5 µM) for the next 48 h. After that, the media was changed and followed by isolation and purification of proteins. Subsequently, Western blot analysis was carried out according to the protocol described above.

### 4.9. Agilent xCELLigence Real-Time Cell Analysis

RAMA 38-28 cells (8 × 10^3^ cells/well) were seeded in 96-well plates (RTCA E-Plates 96) in xCELLigence RTCA systems (Agilent Technologies, Santa Clara, CA, USA) for 24 h followed by the treatment using siRNA (14 nM), or ntRNA (5 nM) for 48 h. Subsequently, cells were treated by PTX (200 nM) for 60 h. The cell adhesion and spread of the cells were continuously monitored in 60 min intervals over the course of monitoring period using the xCELLigence RTCA system.

### 4.10. Statistical Analysis of xCELLigence Analysis

Experiments under all conditions were performed in at least three independent measurements. Mean value and standard deviation were calculated using descriptive statistics. The data were analyzed by using the RTCA software Pro 1.2.1 (ACEA Bioscience, Santa Clara, CA, USA). Statistical analysis was carried out by a non-parametric method, one-way ANOVA using SigmaPlot (Ver. 12.0); *p* < 0.05 was considered significant (see Appendix A).

## 5. Conclusions

We demonstrate for the first time robust 233 DEGs evoked by human EPOR overexpression in rat mammary adenocarcinoma RAMA 37-28 cells. Identified DEGs are associated with many biological processes such as proliferation, apoptosis, tumorigenesis, cellular metabolism, differentiation and others. It is obvious that EPOR overexpression affects also signaling pathways either by attenuation of RAS signaling, small GTPase, cytokine-mediated transduction, intrinsic apoptotic pathway in response to DNA damage and cAMP-mediated signaling or through the activation of STAT5, NOTCH signaling and certain Ephrin and WNT signaling mediators. We are aware of certain limitations of this work, which is based on the comparison of rat EPOR overexpressed RAMA 37-28 cells and their parental control only. However, we believe that our newly prepared EPOR-overexpressed human cell lines will soon confirm EPOR-induced gene expression and its potential tissue-specific origin.

## Figures and Tables

**Figure 1 ijms-24-08482-f001:**
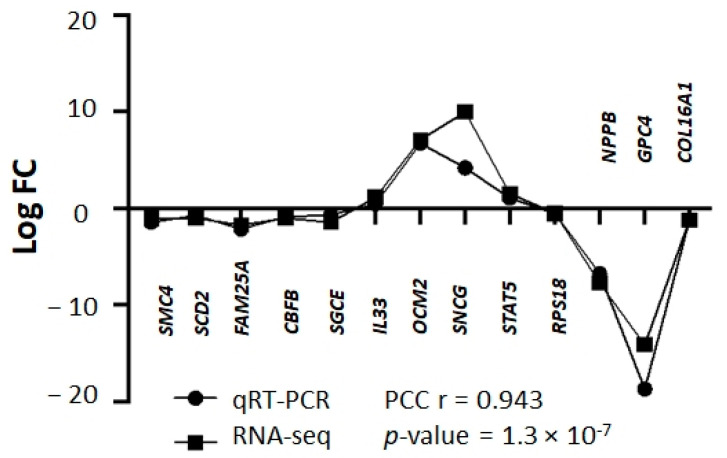
Validation of RNA-seq data using qRT-PCR. Squares bar—logFC values of RNA-seq, bullets bar—logFC values of qRT-PCR. Pearson correlation coefficient (PCC) r = 0.943; *p*-value = 1.3 × 10^−7^. Please note that standard deviation (SD) is not shown here as logFC values of DEGs and qRT-PCR were calculated based on the average CT values of triplicates. The representative results of 13 DEGs are shown.

**Figure 2 ijms-24-08482-f002:**
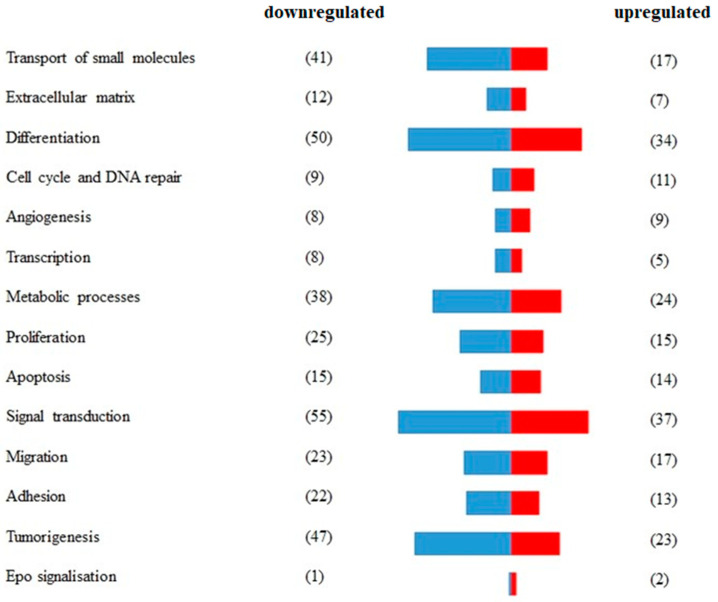
Segregation of differentially expressed genes (DEGs) according to the GO biological processes. A total of 233 DEGs expressed in RAMA 37-28 versus RAMA 37 cells were validated and divided according to GO biological processes and using the peer-reviewed server Omnalysis. Blue bars—downregulated DEGs. Red bars—upregulated DEGs. Number of DEGs is displayed in parenthesis.

**Figure 3 ijms-24-08482-f003:**
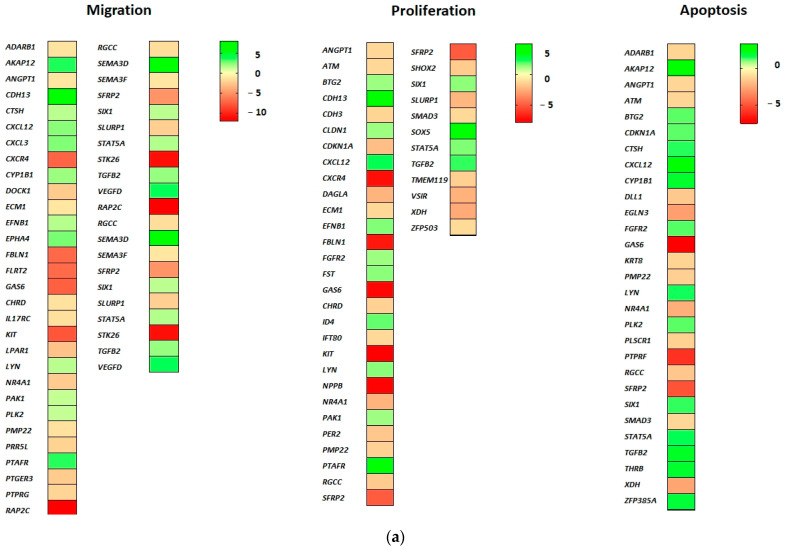
(**a**) DEGs related to proliferation, apoptosis and cell migration. (**b**) DEGs related to metabolic processes and cell differentiation. (**c**) DEGs related to signal transduction, cell transport and tumorigenesis. Green-shaded genes—upregulated, red-shaded genes—downregulated. Shading intensity indicates the degree of upregulation or downregulation. Range of the fold change (logFC values) is presented in the scale.

**Figure 4 ijms-24-08482-f004:**
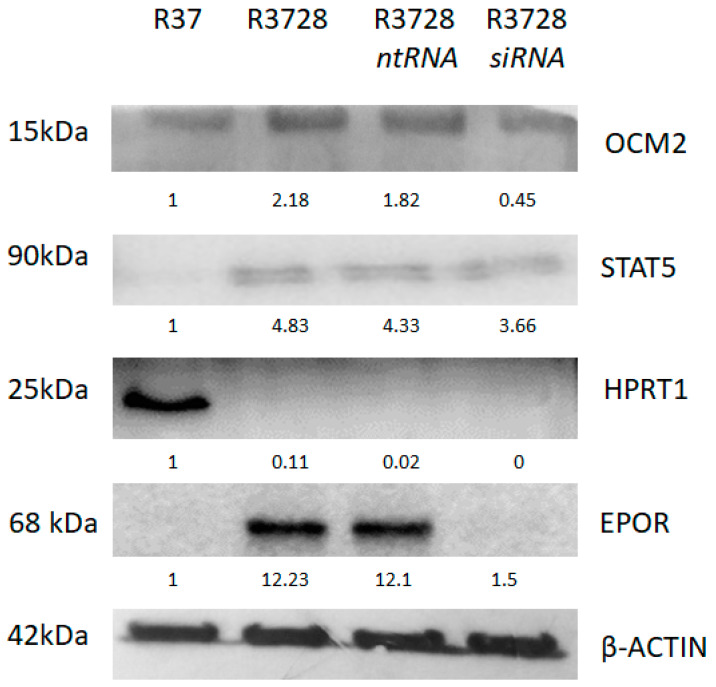
Western blot analysis of representative proteins (OCM2, STAT5A, HPRT1, EPOR) as validation of proteomic data. EPOR-induced changes were confirmed by specific siRNA designed against EPOR mRNA.

## Data Availability

Not applicable.

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
