# Peer review of "Differentially Expressed Genes Induced by Erythropoietin Receptor Overexpression in Rat Mammary Adenocarcinoma RAMA 37-28 Cells"

_ijms, 2023, doi:10.3390/ijms24108482_

Round 1

Reviewer 1 Report

    • The authors based their study on recent finding that EPOR may be associated with metabolic processes, transport of small molecules, signal transduction, and tumorigenesis, and that they are well-known effects on cell proliferation, apoptosis, and differentiation.
    • Their comparative transcriptome analysis (RNA-seq) identified 233 differentially expressed genes (DEGs) in EPOR overexpressed RAMA 37-28 cells compared to parental RAMA 37 cells, with 145 genes downregulated and 88 upregulated.
    • The authors claims to be the first to demonstrate differentially expressed genes evoked by simple EPOR overexpression without the addition of erythropoietin ligand, in a manner that remains to be elucidated.
    • I believe that the manuscript describes sound methodology and good conclusions that researchers in the field will find useful.
  • There are minor errors in the text and grammar such as:
  • progenitor cells instead progenitors
  • insuline should be insulin,
  • embryonic should be used instead of embryonal,
  • signaling instead of signalization,
  • ephrin not ephrine
  • etc.

Author Response

  • The authors based their study on recent finding that EPOR may be associated with metabolic processes, transport of small molecules, signal transduction, and tumorigenesis, and that they are well-known effects on cell proliferation, apoptosis, and differentiation.
  • Their comparative transcriptome analysis (RNA-seq) identified 233 differentially expressed genes (DEGs) in EPOR overexpressed RAMA 37-28 cells compared to parental RAMA 37 cells, with 145 genes downregulated and 88 upregulated.
  • The authors claims to be the first to demonstrate differentially expressed genes evoked by simple EPOR overexpression without the addition of erythropoietin ligand, in a manner that remains to be elucidated.
  • I believe that the manuscript describes sound methodology and good conclusions that researchers in the field will find useful.
  • There are minor errors in the text and grammar such as:
  • progenitor cells instead progenitors                       Corrected
  • insuline should be insulin,                                       Corrected
  • embryonic should be used instead of embryonal,  Corrected
  • signaling instead of signalization,                           Corrected
  • ephrin not ephrine                                                  Corrected

  • Thank you for your comments. We have accepted all of your suggestions and removed mistakes in the manuscript.

Reviewer 2 Report

Manuscript ID: ijms-2343320
Type of manuscript: Article

The manuscript entitled „Differentially expressed genes induced by erythropoietin receptor overexpression in rat mammary adenocarcinoma RAMA 37-28 cells” was submitted to the International Journal of Molecular Sciences, on the topic Cancer Cell Metabolism. The work describes the analysis of the transcriptome (15,177 genes) and proteome (5,000 peptides) from RAMA 37-28 cells in relation to parental RAMA 37. It was observed that overexpression of EPOR in RAMA 37-28 cells caused changes in the expression of 233 genes, which were divided into upregulated and downregulated, and because of their involvement in cellular processes and functions. Of these, 13 were selected and validated by qRT-PCR and compared with the seqRNA results. From the proteome analysis, OCM2, STAT2, HPRT1, and EPOR were selected by validating them with Western blot and siRNA silencing. Overexpression of human EPOR in RAMA 37-28 rat mammary adenocarcinoma cells has been shown to alter the expression of important genes involved in cellular processes such as proliferation, apoptosis, cellular metabolism, differentiation, and multiple signaling and transduction pathways.

My comments and suggestions:

1) Please, change the order of the sections according to the requirements of the IJMS journal.

2) The entire methodology lists companies without a city or country - please complete. In addition, abbreviations mentioned for the first time in the text should be described in parentheses.

3) Gene expression was calculated in the unprotected website of the base 2 logarithm calculator (web address provided). However, it is written that it is a conversion of deltadeltaCt to the common logarithm of FC. How was fold change calculated? Please check and refer to the source article on the methodology of calculating gene expression and FC.

4) There are inaccuracies in the description of "Statistical analysis": descriptive statistics were calculated as mean value and standard deviation, where did you show?  Figure 1 is logFC, Figure 2 number of downregulated and upregulated genes, Figure 3a – c is the division of DEGs due to participation and functions in cellular processes, and Figure 4 is Western blot analysis.

5) When reading the work and the description of the statistical analyzes used in the work, one does not see consistency. In this subchapter, it is necessary to write exactly which analyses were performed with which software. The paper does not divide the results according to the p-value (*, **, ***), so you should write which results are affected.

6) Figure 1 - there should be a space between Log and FC on the OY axis; one time it writes QRT-PCR and the other time qRT-PCR, this applies to all explanations - please harmonize: RNA-SEQ vs. seqRNA; p value vs. P-value.

What is on the OX axis? Genes? Why are they lowercase? They are written in capital letters in the text (genes ought to be written in capital letters and italics).

The title of Figure 1 is probably incorrect, because what are we validating?

 qRT-PCR is a validation method.

The value of the correlation coefficient r proves that the values of A and B were correlated. What values were correlated here? Is it an average value? Of which? Why p value once = 1.3 x 10-7 and below in the description p < 0.01?

There is no consistency in the description of the results and their presentation.

7) Figure 2 - It would be necessary to write downregulated or upregulated above the numbers of genes. Usually, the descriptions of gene expression in warm colors describe an increase, and in cold colors a decrease.

8) Results text:

Please explain why it says:

"r=0.943 for RAMA37-28" ? correlation for cells?

  "After validating ....." validation is carried out after reviewing all genes and based on the criteria, suitable ones for validation are selected, i.e. maybe it should be written: Before validating ..... peer-reviewed server - OMnalysis ...

You write CYP1B1 and CYP1b1 - correct in this case and others.

9) Literature requires correction by the requirements of the IJMS journal.

10) The work has many typo errors, usually made by non-native speakers

Author Response

Thank you for your comments and suggestions, which helped us to improve our manuscript. We tried to remove all the mistakes in the text and incorporate your proposals.

1) Please, change the order of the sections according to the requirements of the IJMS journal.

We have changed order of the sections according to requirements of IJMS.

2) The entire methodology lists companies without a city or country - please complete. In addition, abbreviations mentioned for the first time in the text should be described in parentheses.

Cities and countries were added to the methodology list of companies.

3) Gene expression was calculated in the unprotected website of the base 2 logarithm calculator (web address provided). However, it is written that it is a conversion of deltadeltaCt to the common logarithm of FC. How was fold change calculated? Please check and refer to the source article on the methodology of calculating gene expression and FC.

We apologize for providing unprotected and incorrect website, it’s our faul. We have changed the website (https://goldbio.com/qpcr-and-rtqpcr-analysis-tool) and providing the correct one in the manuscript.

4) There are inaccuracies in the description of "Statistical analysis": descriptive statistics were calculated as mean value and standard deviation, where did you show?  Figure 1 is logFC, Figure 2 number of downregulated and upregulated genes, Figure 3a – c is the division of DEGs due to participation and functions in cellular processes, and Figure 4 is Western blot analysis.

Thank you for this comment. Statistical analysis described in our manuscript is related to xCELLigence analysis (Suppl. Fig S5). We have added this information in the manuscript.

5) When reading the work and the description of the statistical analyzes used in the work, one does not see consistency. In this subchapter, it is necessary to write exactly which analyses were performed with which software. The paper does not divide the results according to the p-value (*, **, ***), so you should write which results are affected.

We apologize again and I hope we have already explained through the previous answer.

6) Figure 1 - there should be a space between Log and FC on the OY axis; one time it writes QRT-PCR and the other time qRT-PCR, this applies to all explanations - please harmonize: RNA-SEQ vs. seqRNA; p value vs. P-value.

Thank you for suggestions, we have unified the terms in the manuscript.

What is on the OX axis? Genes? Why are they lowercase? They are written in capital letters in the text (genes ought to be written in capital letters and italics).

Thank you, we have already changed it.

The title of Figure 1 is probably incorrect, because what are we validating?

The title of Figure 1 was changed  and is as it follows: Validation of RNA-seq data using qRT-PCR.

RT-PCR is a validation method.

The value of the correlation coefficient r proves that the values of A and B were correlated. What values were correlated here? Is it an average value? Of which? Why p value once = 1.3 x 10-7 and below in the description p < 0.01?

The logFC values of RNA-seq and qRT-PCR were correlated in Figure 1. The description under the Figure 1 has been modified and I believe is more understandable.

There is no consistency in the description of the results and their presentation.

We have modified Figure 1 according to your suggestion and I hope the description of the results is more consistent with their presenstation.

7) Figure 2 - It would be necessary to write downregulated or upregulated above the numbers of genes. Usually, the descriptions of gene expression in warm colors describe an increase, and in cold colors a decrease.

Thank you very much. According to your suggestion we have modified Figure 2.

8) Results text:

Please explain why it says:

"r=0.943 for RAMA37-28" ? correlation for cells?

We apologize, but RAMA37-28 were mistakenly mentioned in this section of the manuscript. Indeed, we are still comparing logFC values of RNA-seq and qRT-PCR data.

"After validating ....." validation is carried out after reviewing all genes and based on the criteria, suitable ones for validation are selected, i.e. maybe it should be written: Before validating ..... peer-reviewed server - OMnalysis ...

Accepted.

 You write CYP1B1 and CYP1b1 - correct in this case and others. 

Accepted.

 9) Literature requires correction by the requirements of the IJMS journal.

The requirements of IJMS regarding the literature were checked again and followed.

10) The work has many typo errors, usually made by non-native speakers

Thank you. The manuscript was checked by native english speaker and typo errors corrected.

Reviewer 3 Report

Thank you for giving me the opportunity to review this article.

The authors analyzed gene expression changes related to stable EPOR overexpression using RAMA 37 and RAMA 37-28 cell lines. Related genes were narrowed down by comprehensive gene analysis and, after that, validated by qPCR. Although these results are meaningful to readers, there seem to be several points for improvement. The followings are my suggestions.

Use additional cell lines. This research's results are based only on RAMA 37 and RAMA 37-28 cell lines. And these cell lines are not human-derived. I suggest using additional cell lines to add credibility and evidence to this report. If the authors cannot do this, please describe its difficulty as a limitation in the discussion part.

Author Response

The authors analyzed gene expression changes related to stable EPOR overexpression using RAMA 37 and RAMA 37-28 cell lines. Related genes were narrowed down by comprehensive gene analysis and, after that, validated by qPCR. Although these results are meaningful to readers, there seem to be several points for improvement. The followings are my suggestions.

Use additional cell lines. This research's results are based only on RAMA 37 and RAMA 37-28 cell lines. And these cell lines are not human-derived. I suggest using additional cell lines to add credibility and evidence to this report. If the authors cannot do this, please describe its difficulty as a limitation in the discussion part.

Thank you for your comments and suggestions. As you have expected, there is a problem to add additional cell line immediately, even if we are preparing another human cell lines over-expressed by EPOR. For this manuscript, we have decided to explain two cell lines limit in the conclusions of the manuscript as follows: “We are aware of certain limitations of this work, which is based on the comparison of rat EPOR overexpressed RAMA 37-28 cells and their parental control.However, we believe that our newly prepared EPOR-overexpressed human cell lines will soon confirm EPOR-induced gene expression and its potential tissue-specific origin.

Round 2

Reviewer 3 Report

Thank you very much for sending me the second version. The manuscript has been revised well. I think this revised manuscript is suitable for acceptance.